# Establishing and augmenting views on the acceptability of a paediatric critical care randomised controlled trial (the FEVER trial): a mixed methods study

Elizabeth Deja ,[1] Mark J Peters,[2,3] Imran Khan,[4] Paul R Mouncey,[5] Rachel Agbeko,[6] Blaise Fenn,[7] Jason Watkins,[7] Padmanabhan Ramnarayan,[8] Shane M Tibby,[9] Kentigern Thorburn,[10] Lyvonne N Tume,[11] Kathryn M Rowan,[12] Kerry Woolfall[1]

For numbered affiliations see end of article.

**Correspondence to**
Dr Elizabeth Deja;
e.deja@liverpool.ac.uk

## ABSTRACT

**Objective** To explore parent and staff views on the acceptability of a randomised controlled trial investigating temperature thresholds for antipyretic intervention in critically ill children with fever and infection (the FEVER trial) during a multi-phase pilot study.

**Design** Mixed methods study with data collected at three time points: (1) before, (2) during and (3) after a pilot trial.

**Setting** English, Paediatric Intensive Care Units (PICUs).

**Participants** (1) Pre-pilot trial focus groups with pilot site staff (n=56) and interviews with parents (n=25) whose child had been admitted to PICU in the last 3 years with a fever and suspected infection, (2) Questionnaires with parents of randomised children following pilot trial recruitment (n=48 from 47 families) and (3) post-pilot trial interviews with parents (n=19), focus groups (n=50) and a survey (n=48) with site staff. Analysis drew on Sekhon *et al*'s theoretical framework of acceptability.

**Results** There was initial support for the trial, yet some held concerns regarding the proposed temperature thresholds and not using paracetamol for pain or discomfort. Pre-trial findings informed protocol changes and training, which influenced views on trial acceptability. Staff trained by the FEVER team found the trial more acceptable than those trained by colleagues. Parents and staff found the trial acceptable. Some concerns about pain or discomfort during weaning from ventilation remained.

**Conclusions** Pre-trial findings and pilot trial experience influenced acceptability, providing insight into how challenges may be overcome. We present an adapted theoretical framework of acceptability to inform future trial feasibility studies.

**Trial registration numbers** ISRCTN16022198 and NCT03028818.

## Strengths and limitations of this study

► The longitudinal design enabled collection of data from parents and staff with relevant experience before, during and after the pilot trial.
► The mixed methods approach, including interviews, focus groups and surveys, enabled breadth and depth of insight to help establish trial feasibility.
► Use of the Sekhon *et al*'s theoretical framework of acceptability allowed trial acceptability to be evaluated as a multifaceted construct as opposed to a poorly defined binary (acceptable/not acceptable) approach.
► Data collected during the pilot trial stage were limited to parent perspectives, the majority of whom were mothers, although staff views were sought retrospectively.

conduct due to ethical and practical considerations that are not applicable to trials in adult settings.[4–8] For example, the eligible population is smaller and consent is obtained by proxy through children's parents or legal guardians.[4 9–11] These considerations are compounded in critical care settings by the emotive and time sensitive situation in which they take place. Clinical trials must be acceptable to parents and healthcare practitioners to facilitate recruitment, adherence and consent.[12 13] Sekhon *et al*'s[13] present a theoretical framework of acceptability (TFA) (see figure 1) to assist researchers in assessing the acceptability of healthcare interventions, including clinical trials. The TFA presents seven theoretical constructs for researchers to consider when assessing whether people delivering or receiving a healthcare intervention consider it to be appropriate. The constructs highlight considerations when establishing acceptability, such as how an

## INTRODUCTION

Recruitment and retention in clinical trials is a significant challenge, which leads to underpowered trials and the continued use of healthcare interventions that are not informed by robust scientific evidence.[1–3] Paediatric clinical trials are particularly challenging to

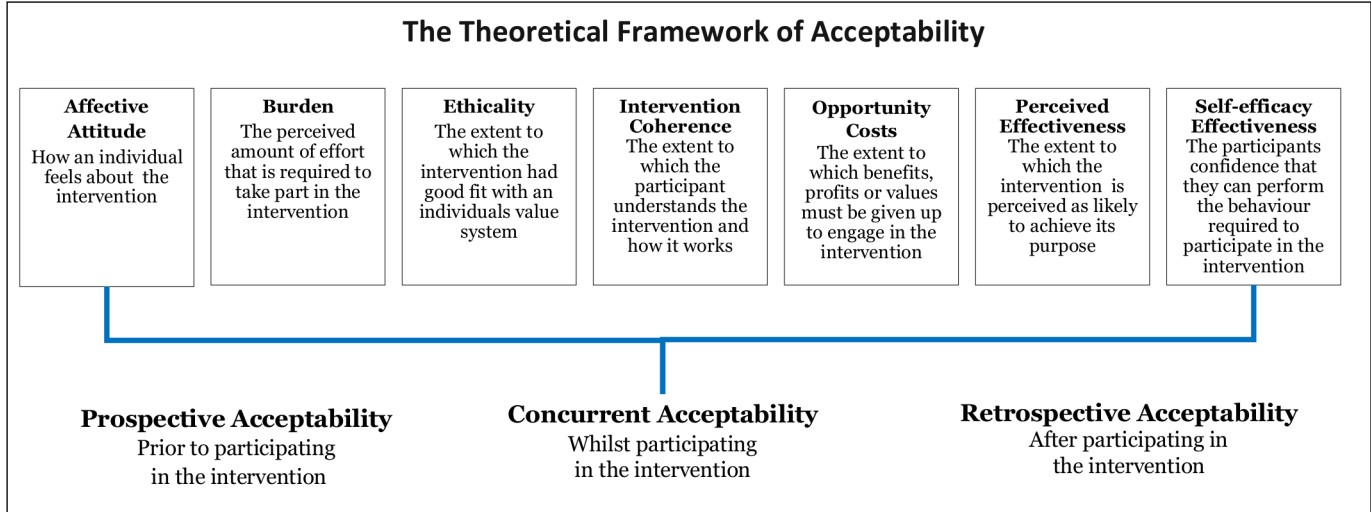

**Figure 1** Sekhon *et al*'s theoretical framework of acceptability.

individual feels about the healthcare intervention, the perceived burden of taking part and the extent to which the participant understands the healthcare intervention and how it works.

There is a lack of robust evidence or guidance to inform the management of fever due to an infection in critically ill children.[14 15] The FEVER feasibility study aimed to establish whether it is possible to conduct a hospital-based clinical trial comparing a permissive approach (treat at ≥40°C) with a restrictive approach (treat at ≥37.5°C) to fever management in children. Perceived challenges to the successful conduct of a fever randomised control trial (RCT) included: a protocol that was likely to differ from usual clinical practice; potential parental and staff concerns about allowing a child's fever to rise without treatment; no time to seek informed consent[16 17] and the possibility that children may die before trial participation is discussed with parents.

The FEVER feasibility study involved a multi-phase pilot study, including pre-trial research with parents and staff, an observation of UK practice and a subsequent pilot RCT with embedded research exploring the perspectives of parents and staff involved in the pilot RCT.[18–20] This paper focuses on research exploring parent and staff perspectives on trial acceptability drawing on the TFA.

## METHODS
### Study design
As part of the wider FEVER feasibility study (see figure 2), we conducted mixed methods research involving interviews, focus groups and surveys with parents who had relevant experience and staff involved in the pilot RCT. The research reported in this paper was conducted before, during and after the pilot RCT and aimed to explore parent and staff views on the proposed FEVER trial, including trial acceptability, design of information materials, temperature thresholds and the use of research without prior consent (RWPC). The pilot RCT

took place over a 3-month period (October–December 2017). Children were randomly allocated (1:1) without prior consent to permissive (39.5°C) or restrictive

---

**Stage 1**

Pre trial feasibility research including:

- Interviews with Parents with experience their child being admitted to an intensive care unit with a fever and suspected infection in the preceding 3 years

- Focus groups with clinicians (nurses and doctors) working the 4 PICUs planned to be included in the pilot

- Observational study of UK practice related to fever management*

---

**Stage 2**

- Pilot RCT in 4 hospitals comparing a permissive approach (treat as ≥40°C) with a restrictive approach (treat at ≥37.5°C) to fever management in children*

- Embedded survey to explore parent perspectives at the point of trial recruitment.

---

**Stage 3**

- Interviews with parents of children randomised to the pilot RCT approximately one month after hospital discharge.

- Survey and focus groups with staff involved in the pilot RCT at the end of trial recruitment.

**Figure 2** Fever feasibility study design. *reported separately. PICUs, Paediatric Intensive Care Units; RCT, randomised control trial.

(37.5°C) temperature thresholds for antipyretics during their Paediatric Intensive Care Unit (PICU) stay while mechanically ventilated.[19] We used previous research that had explored patient and staff perspectives on trials conducted in paediatric emergency and critical care in the NHS[21–23] to develop topic guides for interviews and focus groups, questionnaires and participant information sheets (PIS).

## Patient and public involvement

Two parents (Clara Francis and JW) and one young adult (BF) with experience of severe infection and admission to hospital were co-investigators and members of the Study Management Group. They provided valuable input into the design and conduct of the study, including reviewing documents for parent interviews (eg, draft pilot trial PIS) and informing study recruitment approaches (ie, identification of social media groups and charities). They were also involved in the review of study progress and findings.

## Pre pilot trial: prospective recruitment and conduct
### Parent interviews

English-speaking parents of children (under 16 years) that had been admitted to an intensive care unit with a fever and infection in the last 3 years were recruited via a database from a previous relevant study[24]; a letter from study sites[19] and advertising on relevant social media and at sites. Some leniency was allowed if the child was admitted close to 3 years prior to interview (eg, 3 years and 2 months). All routes invited parents to register interest in participation by contacting the research team.

Psychologist ED (PhD, female research associate) responded to parents' requests to participate in sequential order and checked eligibility. A draft pilot RCT PIS was emailed to parents prior to interview, which included an outline of the study and current practice on the management of fever in critically ill children. ED conducted interviews in person or via telephone based on parent preference. Audio-recorded verbal or written consent was sought before interviews as appropriate. Audio consent involved reading each aspect of the consent form to parents, including consent for audio recording and to receive a copy of the findings when the study is complete. Each box was initialled on the consent form when verbal consent was provided. Informed consent discussions were audio recorded for auditing purposes. Interviewing stopped when data saturation[25] and variation in sample was reached.

### Staff focus groups

Co-investigators at the four pilot RCT sites[19] disseminated invitations to all staff who would be involved in the conduct of a clinical trial within a PICU. KW (PhD, female, social scientist and a senior lecturer) or ED provided a PIS and obtained written informed consent before the focus group began. The topic guide consisted of a mix of open-ended and closed-ended questions. Closed-ended questions were administered using the Turning Technologies (Youngstown, Ohio, USA) voting system. This allowed for the collection of staff demographic information, to ensure data collection from all staff on key questions, such as views on trial acceptability. The use of Turning Point also enabled us to show grouped findings for closed-ended questions on a screen to explore reasons for views in more depth verbally during the discussion.

## Pilot RCT: concurrent and retrospective recruitment and conduct

Interim analysis of prospective data informed subsequent topic guides and questionnaires.

### Parent questionnaire and interviews

As part of the pilot RCT consent discussions, site researchers asked both parents if they would like to complete the FEVER consent questionnaire after the pilot RCT recruitment discussion (concurrent) and/or take part in a telephone interview approximately a month later (retrospective). In addition to collecting minimal demographic information, the consent questionnaire asked them to indicate how strongly they agreed or disagreed with 12 statements about the FEVER RCT followed by tick box and open-ended responses regarding their consent decision. ED contacted those who consented to interview in sequential order (by receipt of a consent form), stratifying by study arm (lower/higher temperature threshold) as the study progressed ensuring parents whose children had been randomised to both trial arms were represented in the sample.

### Staff survey and focus groups

At the end of the pilot RCT, ED repeated focus groups with staff at the four pilot RCT sites to explore their experiences of pilot trial conduct and views on the proposed trial acceptability. Those unable to attend a focus group were invited by email to complete an online questionnaire containing the same closed-ended questions administered to focus group participants using the Turning Technologies voting system.

### Analysis

Digital audio recordings were transcribed verbatim by a professional transcription company (Voicescript Ltd, Bristol, UK). Transcripts were anonymised and checked for accuracy.

ED and KW used a thematic analysis approach[26] to explore themes within the data related to views on trial design and acceptability (see table 1). Analysis was interpretive and iterative.[26 27] NVivo V.10 software (QSR International Pty Ltd, Melbourne, Australia) was used to assist in the organisation and coding of qualitative data. Quantitative data from the parent and staff questionnaires were entered into SPSS V.20.0 and analysed using descriptive statistics. Please see separate publication for further details.[19] ED and KW then synthesised data and used framework analysis[28] to map findings onto each component of the TFA by time point[13] (see tables 2–4). Where

| Table 1 | Approach to thematic qualitative data analysis |
|---|---|
| **Phase** | **Description** |
| 1. Familiarising with data | ED and KW read and re-read transcripts noting down initial ideas on themes. |
| 2. Generating initial codes | Two complementary data-coding frameworks were developed (one focus group data (KW) and one interview data (ED)) using a priori codes identified from the project proposal and topic guilds. During the familiarisation stage ED and KW identified additional data-driven codes and concepts not previously captured in the initial coding frame. |
| 3. Developing the coding framework | KW coded 10% of the interview transcripts using the initial coding frame and made notes on any new themes identified and how the framework could be refined. In turn, ED coded 10% of the focus group transcripts following the same procedure. |
| 4. Defining and naming themes | Following review and reconciliation revised coding frames were subsequently developed and ordered into themes. |
| 5. Completion of coding of transcripts | ED completed coding interview transcripts and KW completed coding focus group transcripts in preparation for write-up. |
| 6. Producing the report | ED and KW developed the original manuscript using themes to relate back to the study aims ensuring key findings and recommendations were relevant to the FEVER trial design and site staff training (ie, catalytic validity). Final discussion and development of selected themes occurred during the write-up phase. |

illustrative quotes are provided, the participant identifier relates to each participant (eg, P01 is participant 1).

## RESULTS
### Participant characteristics
Prospective (pre-pilot RCT): 25 semi-structured interviews (n=20 mothers, n=5 fathers) with bereaved (n=6) and non-bereaved (n=19) parents (see figure 3). Parents were interviewed a median of 14 months (range: 6–38 months) after admission. Interviews took a median of 48 min (range: 15–105 min). The 15-minute interview was concluded part way through by a bereaved father. Fifty-six staff took part in six focus groups across the four sites, lasting a median of 50 min (range: 31–59 min). Staff mainly self-identified as nurses (n=45, 81%) were involved in the clinical care of children.

Concurrent (during-pilot RCT): 80 parents of the 100 children randomised to the pilot RCT consented to receive questionnaire, of these, 48 from 47 families completed and returned a questionnaire while their child was admitted to hospital. Of these, 41/48 (85%) provided consent and 6/48 (13%) declined consent (n=1 missing).

Retrospective (post-pilot RCT): 66 parents of the 100 children randomised to the FEVER pilot trial consented to be contacted for an interview. Data saturation[25] was reached after 8 interviews with parents of children allocated to the restrictive (lower) temperature threshold and after 11 interviews with parents of children allocated to the permissive (higher) temperature threshold. Parents were interviewed a median of 31 days after randomisation (range: 9–70 days). Their children had received treatment for respiratory illness (eg, bronchiolitis and respiratory syncytial virus) (n=18/19, 94%), cancer (n=1/19, 5%) and septic shock (n=1/19, 5%). Interviews took an average of 32 min (range: 20–50 min).

The staff sample included 98 site staff across all four pilot RCT sites. Almost half (48/98, 49 %) completed the questionnaire, with the rest attending a focus group. The majority (n=75, 77%) were nurses, n=45 (60%), were senior-level staff and most (n=79/98, 81%) were involved in the clinical care of children. Focus groups took an average of 53 min (range: 23–106 min).

### Pre-trial, prospective acceptability
All parents interviewed described how they would hypothetically consent for the use of their child's information in the proposed trial. Parents' views on trial acceptability appeared to be influenced by factors, including all other treatments for infection are given (opportunity costs); the non-invasive nature of the intervention (burden); support for RWPC in this context (ethicality); trust in medical staff to act in the best interests of their child and a belief that the trial question made sense and, therefore, likely to achieve its purpose (perceived effectiveness):

'cause fever is meant to be like part of a fighting off, healing process isn't it? A natural one… I can understand exactly why it would be interesting to see what happens.' (P07, mother, non-bereaved)

Although analysis of pre-RCT data indicated that many of the constructs of acceptability were met (see table 2), there were also aspects of burden, opportunity costs, ethicality and intervention coherence identified as problematic by both parents and staff. As listed in table 2, staff concerns outweighed support at this stage.

The majority of parents were not worried about the proposed restrictive temperature threshold of 37.5°C. However, staff expressed concerns that this was too low a threshold to administer an antipyretic (n=43/54, 80%, two missing) and would go against perceived 'normal practice' (P01, Staff, FG4). A common concern was that

**Table 2** Prospective acceptability of a FEVER pilot trial mapped to Sekhon et al's acceptability framework

| Group and data collection method | Affective attitude | Burden | Ethicality | Intervention coherence | Opportunity costs | Perceived effectiveness | Self-efficacy |
|---|---|---|---|---|---|---|---|
| Parents Interviews | 100% stated they would consent for their child to take part in a FEVER RCT. Would consent with a 40°C threshold, but 39.5°C–39.9°C more acceptable: 'I think 39.5(°C). But again you guys know best I'm just saying …that's very hot'. (P17, father, non-bereaved) | The intervention was not invasive | Belief; it is important to help other children in the future. Use of RWPC is necessary: 'I understand there's not really another way you can do it' (P01, mother, non-bereaved) | Logically 'make sense' (P02, mother, bereaved). PIS: 'it's simple for them to read'. (P06, mother, bereaved) | Children would still be given all other kinds of care/interventions. | When study rationale was explained parents understood how allowing a fever could have a positive impact: 'Fever is meant to be like part of a fighting off, healing process isn't it?'. (P07, mother, non-bereaved). | The intervention was something parents understood and said that they could support. Important to approach for RWPC when parents have the capacity to make an informed decision. |
| | Concerns about unnecessary discomfort/pain in higher threshold. | | | Many suggested changes to the PIS to assist understanding and decision-making. | Concerns about loss of non-antipyretics effects of paracetamol, eg, reducing risk of seizures/rigours and pain relief. | | |
| Staff Focus group | 82% (45/55, one missing) indicated 39.5°C was an acceptable permissive temperature threshold. 18.2% suggested 40°C was acceptable. Only 20.4% suggested 37.5°C was acceptable as may lead to unnecessary intervention. | Watching a child be in pain or experience negative side effects: 'Incredibly difficult to wait and watch'. (P05, Staff, FG5). The trial would be more acceptable if limited to ventilated children. | Mixed views on RWPC, n=25/49, (51%) thought acceptable based on past experience and the emergency situation. Concerns about use of RWPC for an intervention that may not be supported by parents. | Understanding that optimal temperature thresholds are unknown | Concerns about the loss of non-antipyretic effects for example, discomfort relief, reducing risk of seizures/rigours, decreased cardio work load. | Evidence to support the trial: 'Well there is, there is a bit of science which suggests we should let the temperature get higher'. (P01, Staff, FG3) | No perceived issues with taking a temperature. Query method that is going to be used. |
| | | | | Want more clinical evidence as it goes against experiential knowledge (eg, administering antipyretic at 38°C). | Staff with no experience of RWPC had concerns it would negatively impact on trust and the 'working relationship'. (P03, Staff, FG1) | Waiting for the permissive threshold would go against their clinical training or 'gut instinct'. (P05, Staff, FG2) | Nurses stated that they may not follow the protocol if a child was upset, combative and in discomfort. |

Key: shaded fields highlight potentially unacceptable aspects of the trial.
FG, Focus group; PIS, participant information sheets; RCT, randomised control trial; RWPC, research without prior consent.

**Table 3** Parent concordant acceptability of a FEVER pilot trial mapped to Sekhon et al's acceptability framework

| Data collection method | Affective attitude | Burden | Ethicality | Intervention coherence | Opportunity costs | Perceived effectiveness | Self-efficacy |
|---|---|---|---|---|---|---|---|
| Parents<br>Questionnaire | N=92/100 (92%) consented to taking part in the FEVER pilot RCT. | Not collected at this time point. | N=32/41 (79%) reported the belief that medical research studies are important.<br>N=42/48 (89.4%) satisfied with the RWPC process in the FEVER pilot RCT.<br>N=32/41 (78%) selected helping other children as a reason for taking part. | N=46/48 (96%) agreed that the information received about the FEVER pilot RCT was clear and straightforward to understand. | 'My child is comfortable' (P49, questionnaire, mother, permissive)<br><br>Concerns about their child being in pain or discomfort and impact on pre-existing medical condition. 'My son had too many underlying medical conditions and felt it may hinder his recovery as he was selected to the upper limit before treatment.' (P73, questionnaire, father, permissive) | N=30/41 (73.2%) selected helping my child as a reason for taking part: 'So far recovering well'. P21,mother,non-bereaved) | N=41(89.4%) felt hat they made the decision for their child to take part in the pilot trial. |

Key: shaded fields highlight potentially unacceptable aspects of the trial.
RCT, randomised control trial; RWPC, research without prior consent.

children would be given unnecessary treatments in a clinical context where 'we try and give the minimum amount of drugs' (P02, Staff, FG5) (affective attitude).

In contrast, many parents voiced concerns about the acceptability of the permissive threshold with regards to increased risk of 'seizures' (P03, mother, non-bereaved) and other potential detrimental side effects (opportunity costs), such as 'organs shutting down' (P07, mother, non-bereaved), 'rigour' (P06, mother, bereaved) or unnecessary discomfort (burden). The majority suggested that the pilot RCT would be more acceptable if the permissive temperature threshold was slightly lower (eg, 39.9°C or 39.5°C). Although parents stated that they would still consent to a trial involving a threshold of 40°C (affective attitude), as they trusted staff to monitor their child and act in their best interests:

'I would trust that my child was being monitored, it's not like they're waiting for her condition to get worse before they do something, you are having, a nurse by your bedside at all times, I had complete trust.' (P25, mother, non-bereaved)

Staff also described how a permissive threshold of ≥40°C was too high and how they would be concerned about not using paracetamol for analgesia in the less unwell, spontaneously breathing patients, who may be in pain or discomfort (burden). In addition, staff were concerned about parental acceptability of the permissive threshold, RWPC (ethicality and opportunity costs) and the impact of increased cardiac workload (opportunity costs).

Both groups understood the aims of the proposed trial. However, in addition to changes to temperature thresholds, amendments to the protocol were suggested. Parents identified aspects of the PIS that required clarification, including whether not treating a temperature could cause a seizure, incorporating an explanation of how all other treatments would still be given. Staff requested additional information about the scientific evidence underpinning the research question, as well as clarification on key issues, such as what cooling methods could be used.

### Response to pre-trial findings

Pre-trial findings were used in conjunction with observation study findings[19] to develop the pilot RCT protocol and site training. These included a permissive temperature threshold of ≥39.5°C; inclusion criteria that required patients to be mechanically ventilated, therefore, likely to be on other analgesia and changes to information materials. For example, staff training and PIS incorporated evidence to demonstrate how fever does not cause seizures and observation study findings that showed the restrictive temperature threshold (≥37.5°C) falls within usual practice. To address staff concerns about how parents may respond to trial and RWPC discussions, parent perspectives were communicated in site training, highlighting parental acceptability of RWPC, temperature thresholds,

**Table 4** Retrospective acceptability of a FEVER pilot trial mapped to Sekhon *et al*'s acceptability framework

| Group and data collection method(s) | Affective attitude | Burden | Ethicality | Intervention coherence | Opportunity costs | Perceived effectiveness | Self-efficacy |
|---|---|---|---|---|---|---|---|
| Parents Interviews | 'I think it's a brilliant idea, so I'm all, I'm all for it' (P80, interview mother, permissive). Decliners also approve the trial 'They seem to be quick let's give them paracetamol. we've always been like a kind of, a hold off (...) I think sometimes paracetamol can hide other things going on as well.' (P83, interview, mother, permissive, decliner) | Not invasive: 'I mean if it was more of an invasive study, I might have had to query it a little bit more but I was happy with everything' (Parent 53, interview, father, restrictive). Not in discomfort while ventilated. When being weaned of ventilator Some children displayed discomfort/distress. | Not causing harm to their child: 'I just thought, There's no harm in letting her into the study' (P78, interview, mother, restrictive). Wish to help others 'if it's gonna help other people then yeah' (P85, interview, father, restrictive). RWPC was acceptable. | Involved treatment which parents were familiar with; that logically made sense and that was 'Clearly explained.' (P74, interview, mother, permissive) | 37.5°C, very acceptable as normal practice at home. 39.5°C acceptable if child is not in discomfort ≥39.5°C not acceptable if the child is in pain or distress. | Understand how a fever may logically help a child. Concerns about negative impact due to child's pre-existing medical conditions. | Valued ability to withdraw or decline consent. 'I was happy enough for him to undergo the trial but if at any point the nurses thought he could do with the Calpol, or I thought he could, then I wanted the trial to stop it could do." (P49, interview, mother, permissive). |
| Staff Focus group and survey | 85% (n=81/95) trial acceptable. 95% (n=95/100, four missing): 37°C acceptable. 53% (n=42/79): 39.5°C acceptable. | If the child is not conscious or not in pain no burden. | Parents think the trial is acceptable; therefore, it is ethical to randomise their children. 89.4% (n=42/48) satisfied with the use of RWPC. | If trained by trial team, then 39.5°C acceptable. | Mixed views, some observed no negative side effects and reported no costs. | Seeing as believing, 'It's a really good, valuable study to see on a larger scale.' (P04, Staff, FG5) | Valued being involved in trial design. Ability to follow the protocol mixed, n=52/96 (54%): 'technically very easy to follow.' (P06, Staff, FG1) |

Key: shaded fields highlight potentially unacceptable aspects of the trial.
FG, Focus group; RCT, randomised control trial; RWPC, research without prior consent.

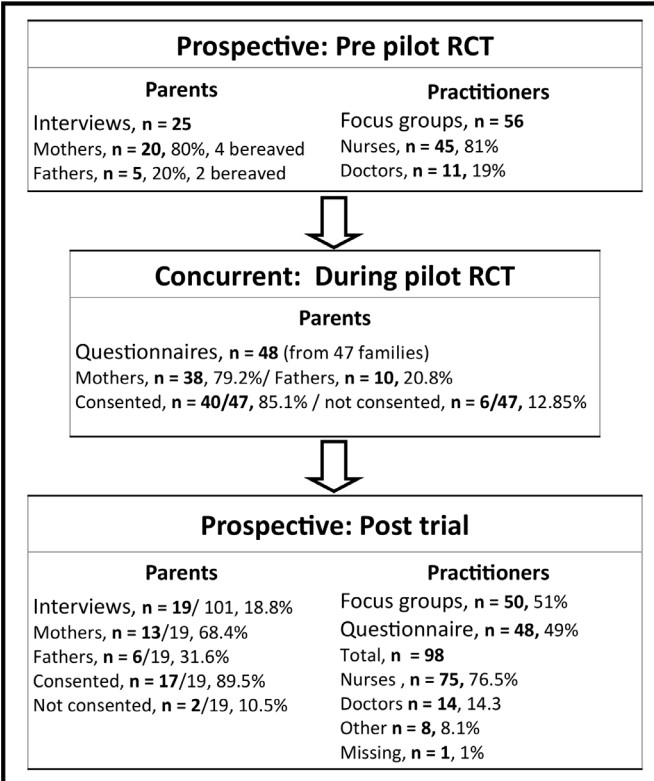

**Figure 3** Participant characteristics by time point. RCT, randomised control trial.

parents' questions about the study and suggestions on how to address such questions.

## Concurrent acceptability

As demonstrated in table 3, parent questionnaire data showed that the six constructs of acceptability measured during this time point were met. Parents reported that the study information 'was clear and straightforward to understand' (intervention coherence). Ninety-two per cent of randomised children received consent for their continued participation in the pilot RCT (affective attitude).[19] Main reasons for providing consent related to the belief that participation might help their child (n=30/41, 73%, perceived effectiveness) and help other children in the future (n=32/41, 78%, ethicality). Parents also found the study acceptable because 'my child was comfortable' (P49, questionnaire, mother, permissive) (opportunity costs and burden).

Of the eight that refused consent to continue, seven (88%) had been allocated to the permissive (higher) treatment group. Data suggested that parents who declined some element of their child participation still supported the proposed FEVER RCT. Reasons for refusal of consent were linked to pre-existing medical conditions and the wish to limit any discomfort experienced by their child: 'My son had too many underlying medical conditions and felt it may hinder his recovery as he was selected to the upper limit' (P73, questionnaire, father, permissive). Suggesting that there were still some concerns about withholding analgesia (opportunity costs).

Staff acceptability was not measured at this point.

## Post-trial, retrospective acceptability

Interviews conducted with parents a median 1-month post-randomisation supported and provided further insights into questionnaire findings. All seven constructs of acceptability were met (see table 4). Parents were interested in the trial question and felt the proposed trial was important (affective attitude and perceived effectiveness). Parents described staff as approaching them appropriately, with well-timed, clear, comprehensive study information leading to strong intervention coherence:

'I understood what they were saying and was happy to sort of go ahead, with the trial. If it wasn't explained to me too well, I probably wouldn't have bothered doing it.' (P77, interview, father, restrictive) (intervention coherence)

Parents of children allocated to the restrictive temperature threshold found the trial very acceptable as giving paracetamol at this temperature was 'something that I would do myself anyway' (P82, interview, father, restrictive) (ethicality). Parents also viewed the permissive threshold to be acceptable. However, this acceptability was conditional on their child not being in discomfort (pportunity costs):

'The only thing would be if she wasn't on any other kind of pain relief, but there's other things to manage, her discomfort.' (P73, interview, father, permissive)

Indeed, two mothers described how they found the trial acceptable and gave full consent, but later chose to withdraw their child from the study when they were being weaned from ventilation and sedation due to concerns about their child being in pain or distress. Parents valued the ability to withdraw or decline consent (self-efficacy). They also described how they trusted staff to act in their child's best interests, including not adhering to the protocol by administering an antipyretic if at any point staff felt that it was needed (burden and opportunity costs):

'I know if anything did happen, you's can stop at any time. Stop it if they saw it was getting out of hand and he, and I felt like it, it wasn't helping, that I would stop it.…they wouldn't let him go to the stage of him getting poorly.' (P85, interview, father, restrictive)

Unlike parents' views, which largely remained consistent across study time points, staff perceptions of the acceptability of the lower temperature shifted during the course of the pilot RCT. Witnessing patient's positive reactions to RWPC and trial discussions and an awareness that ≥37.5°C was usual practice, resulted in 95% (n=95/100, four missing) of staff rating the restrictive threshold as acceptable or very acceptable: 'Everybody that was in the lower end of it, I found were like happy to take part' (P01, Staff, FG4).

Staff had mixed views about the acceptability of the permissive temperature threshold. Approximately half (n=42/79, 53%) indicated that the ≥39.5°C threshold was acceptable. They valued how the trial team responded to their pre-trial concerns by changing the inclusion criteria to omit non-ventilated children (self-efficacy). Some stated that their previous concerns about high temperatures causing harm or discomfort (opportunity costs) and parents having a negative response to the trial and RWPC (ethicality) were not observed:

'Some patients are randomised into the higher temperature and people see that they're actually manageable and it doesn't cause them any harm… It's kind of seeing is believing.' (P03, Staff, FG1)

Staff who did not find the permissive temperature acceptable were concerned about not giving paracetamol for pain or discomfort when a child was conscious (opportunity costs). These concerns meant that some staff administered paracetamol before a child's temperature had reached ≥39.5°C: 'I feel like potentially we're making our patients more uncomfortable' (P01, Staff, FG2).

Interestingly, staff trained by their local unit colleagues were significantly more likely to find the permissive threshold not acceptable when compared with those trained directly by the pilot trial team ($\chi^2(2)=8.78$, p=0.012). Staff trained colleagues also rated site training as being poor (n=11/97, 11%). These staff remained unclear about the scientific rationale for the study and had lower intervention coherence.

Despite issues with aspects of intervention coherence and opportunity costs, overall staff rated the fever trial acceptable (n=81/95, 85%, affective attitude) and practicable to conduct (n=80/95, 84%, self-efficacy). Findings suggest that their views could be further augmented if the proposed FEVER RCT protocol was revised to also exclude patients receiving non-invasive forms of ventilation (eg,

high-flow nasal oxygen) or those close to being extubated when sedation is being weaned.

## Trust

During data analysis, we found that the concept of trust between parents and staff was prevalent within our data and intrinsically linked to trial acceptability. For example, parents found the trial acceptable because they trusted staff to put the needs of their child before the requirements of the study. Both groups discussed the trust parents place in medical expertise during a very emotive situation. Staff also highlighted that maintaining parental trust impacts on their decisions: 'I feel like there's an element of trust there that would be broken from my point of view' (P01, Staff, FG2, retrospective). The construct of 'Trust' is not reflected within the TFA. We present an adapted TFA in figure 4 incorporating Trust.

## DISCUSSION

Our study highlights the value of conducting pre-trial research with key stakeholders to inform the design of challenging clinical trials.[29] Research with parents and staff helped establish trial acceptability, as well as influence and changed perspectives over time. Prospective qualitative research identified mixed staff views, while parents found the trial broadly acceptable. Both the parental and staff support for RWPC in time critical trials is constant with previous research.[24 30–34] Aspects of intervention coherence, opportunity costs, ethicality and burden[13] were identified that threatened trial success. The majority of staff concerns related to not using paracetamol or active cooling for pain relief, or to prevent febrile seizures.[18] Prospective findings informed changes to the PIS, staff training package and the addition of mechanical ventilation to inclusion criteria. Data from the concurrent and retrospective time points showed a positive response to such changes, particularly among

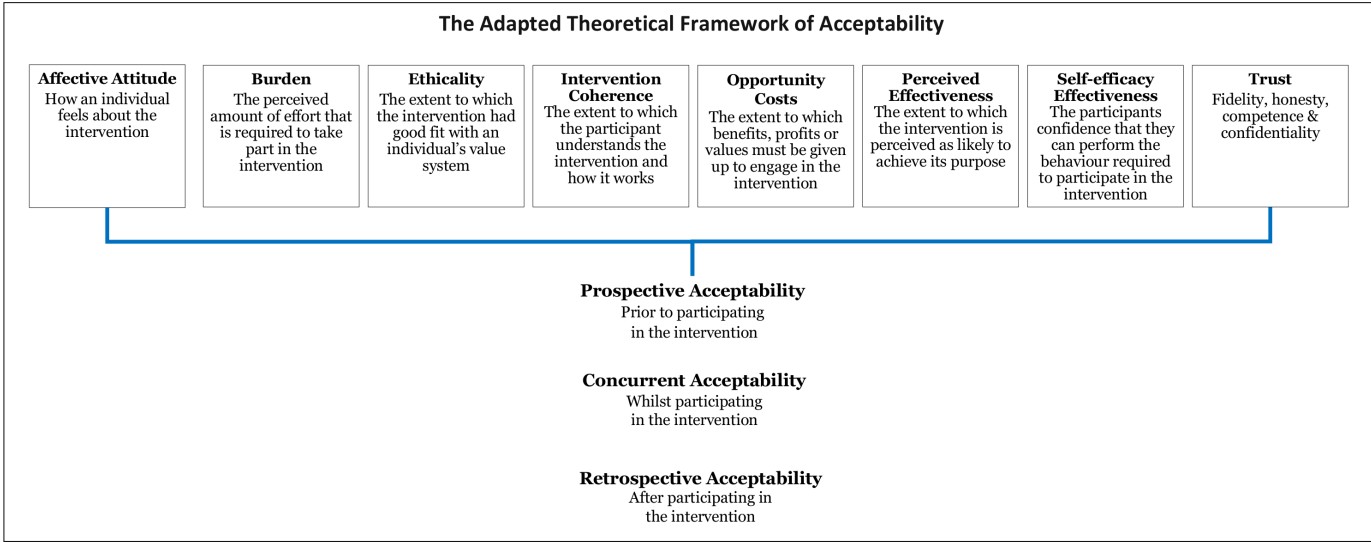

**Figure 4** The adapted theoretical framework of acceptability.

staff. Suggestions to further augment views on trial acceptability and reduce the number of potential protocol deviations and withdrawals were identified. These include changes to trial inclusion criteria as well as staff training content and delivery.[18]

Our findings demonstrate Sekhon *et al*'s assertion that the acceptability of healthcare interventions is not a fixed construct. If we had taken a binary (acceptable/not acceptable) or snapshot approach to determining acceptability, then we would not have been able to identify and address key concerns that threatened trial acceptability and ultimately, trial feasibility. The TFA was demonstrated to be comprehensive and relevant to our work. However, we found that the concept of trust between parents and staff was closely linked to trial acceptability and is not reflected in the framework. The importance of trust is a recurring theme in healthcare and medicine but is particularly salient in paediatric trials, as the more vulnerable the population, the greater the need for trust.[5 35] Drawing on Hall *et al*'s[35] work into defining trust in medical relationships, we propose the addition of an eighth construct of 'Trust' to help inform future trial feasibility research (see figure 4). Further research is needed to test the adapted model in establishing the feasibility of other healthcare interventions and settings. This work will help to establish the appropriateness of trust as additional construct in the TFA.

As the pilot trial was conducted in 3 months, during the busy winter period, the concurrent work only included parents and, therefore, lacks insight into staff perspectives during pilot trial conduct. This limitation was compensated for by the use of retrospective (1 week–1 month) mixed methods ensuring a larger sample, through the survey and depth of insights, through focus groups. Insight was gained into the views of 8 (2 interviews, six questionnaires) out of 18 parents (44%) who had declined their child's continued participation in one or more aspect of the pilot RCT. In particular, the interviews with parents who declined consent and nursing staff who found the protocol challenging to follow provided valuable information to assist with refining the study process for a definitive RCT. However, it is unknown whether or not the predominantly positive views of the declining parents who took part in an interview or questionnaire were shared by other parents who declined the FEVER pilot RCT.

In summary, challenges to delivering the proposed trial included staff and parent concerns about the acceptability of the proposed protocol. Pre-trial research, staff training and experience of pilot trial conduct augmented views, providing insight into how challenges may be overcome, such as changes to the inclusion criteria and delivery of site training. We present an adapted TFA to inform the design of future trial feasibility studies.

**Author affiliations**
[1]Public Health, Policy and Systems, University of Liverpool, Liverpool, UK
[2]Paediatric Intensive Care Unit, Great Ormond Street Hospital for Children NHS Trust, London, UK
[3]Infection, Immunity and Inflammation, Institute of Child Health, University College London, London, UK
[4]Institute of Population Health Sciences, Queen Mary University of London, London, UK
[5]Clinical Trials Unit, Intensive Care National Audit and Research Centre, London, UK
[6]Paediatric Intensive Care Unit, Great North Children's Hospital, Newcastle Upon Tyne, UK
[7]Patient partner, London, UK
[8]Children's Acute Transport Service, Great Ormond Street Hospital for Children, London, UK
[9]Paediatric Intensive Care Unit, Evelina London Children's Hospital, London, UK
[10]Paediatric Intensive Care Unit, Alder Hey Children's Hospital, Liverpool, UK
[11]School of Health and Society, University of Salford, Salford, UK
[12]Intensive Care National Audit and Research Centre, London, UK

**Acknowledgements** The authors would like to thank all the parents who shared their experiences with us; their contribution to the research is invaluable. The authors are grateful to all the staff at participating UK hospitals and charities/support groups for their help with recruitment and the UK PICS Study Group (PICS-SG) for their contributions to this study. We would also like to thank Ms Clara Francis as co-investigator and patient partner.

**Contributors** ED conducted the mixed methods research, co-analysed the data, drafted the initial manuscript, and reviewed and revised the manuscript. MJP is the chief investigator of the FEVER study, conceived, designed and oversaw conduct of the FEVER study and critically reviewed the manuscript. IK was the study manager at the ICNARC CTU responsible for day to day FEVER study management and critically reviewed the manuscript. PRM is the head of research at ICNARC CTU, a co-applicant, involved in the design and coordination of the FEVER study, contributed to and reviewed the manuscript. BF and JW were study co-applicants, patient and parent representatives who contributed to the design and conduct of the FEVER study, including mixed methods perspectives elements. PR, a co-applicant, helped design and conduct the FEVER study. RA, SMT and LNT were co-applicants, site principal investigators and critically reviewed the manuscript. KT was the site principal investigator and critically reviewed the manuscript. KMR is the director at ICNARC CTU, study co-applicant involved in designing and overseeing the FEVER study. KW was a co-applicant on the FEVER study, designed the mixed methods perspectives elements of the FEVER study, supervised the mixed methods research, co-analysed the data and reviewed and revised the manuscript.

**Funding** This study was funded by the UK National Institute for Health Research Health Technology Assessment programme (15/44/01) and supported by the National Institute for Health Research, Great Ormond Street Hospital Biomedical Research Centre. The trial sponsor was Great Ormond Street Hospital NHS Foundation Trust, Joint R&D Office GOSH/ICH, 30 Guilford Street, London WC1N 1EH, UK (Research.Governance@gosh.nhs.uk). The Clinical Trials Unit for the FEVER study was the ICNARC CTU. The views expressed in this publication are those of the authors and not necessarily of the funder, the sponsor, the NHS, the NIHR or the UK Department of Health.

**Competing interests** MJP is a member of the NIHR HTA Board. KMR is a member of the NIHR HS & DR Board.

**Patient and public involvement** Patients and/or the public were involved in the design, or conduct, or reporting, or dissemination plans of this research. Refer to the Methods section for further details.

**Patient consent for publication** Not required.

**Ethics approval** Ethical approval for the study was provided by the National Research Ethics Committee (17/LO/1139). Management approvals were obtained from all study sites.

**Provenance and peer review** Not commissioned; externally peer reviewed.

**Data availability statement** No data are available. The datasets generated during and/or analysed during the current study are not publicly available as consent was not sought for data sharing.

**ORCID iD**
Elizabeth Deja http://orcid.org/0000-0002-3626-4927

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
