## [Reviewer comments · BMJ Open]

ARTICLE DETAILS

TITLE (PROVISIONAL)	Establishing and augmenting views on the acceptability of a paediatric critical care randomised controlled trial (FEVER trial): a mixed methods study
AUTHORS	Deja, Elizabeth; Peters, Mark; Khan, Imran; Mouncey, Paul; Agbeko, Rachel; Fenn, Blaise; Watkins, Jason; Ramnarayan, Padmanabhan; Tibby, Shane; Thorburn, Kentigern; Tume, Lyvonne; Rowan, Kathryn; Woolfall, Kerry

VERSION 1 – REVIEW

REVIEWER	Fabrice Ruiz ClinSearch France
REVIEW RETURNED	14-Sep-2020

GENERAL COMMENTS	This article addresses the critical issue of the acceptability of pediatric RCT to parents and health care professionals based on the development of the protocol and appendices (eg: Participant Information Sheets) and the pilot of the FEVER study (protocol for temperature management in critically ill children with fever and infection). Study findings have already been published, and this paper tends to focus on parent and staff perspectives on trial acceptability. General comment Upon reading the abstract, we can thought that the subject of fever management had been chosen by the researchers to serve as a basis for demonstrating the characteristics of their method to increase the preparedness of patients/parents and healthcare professionals for RCT to establish and augment their acceptability. As such, I would suggest clarifying this point in the abstract and introducing it earlier in the introduction section. The authors have certainly put a lot of effort into these investigations, which in my opinion deserve an elaborate illustration of the design in the section "Methods". This could guide the reader to approach the paper more comfortably. Introduction page 4 The first part of the introduction could be enriched with some more recent references: the authors will be able to rely on a document drafted by the Enpr-EMA which has just been edited and available via the following link: https://www.ema.europa.eu/en/documents/other/preparedness-medicines-clinical-trials-paediatrics-recommendations-enpr-ema-working-group-trial_en.pdf
--

	Line 32-33 “To our knowledge, the TFA has not been used in the analysis of real world data”. This sentence should be reworded because there are now a few references in recent literature. Line 38-40 “There is a lack of robust evidence or guidance to inform the management of fever due to an infection in critically ill children.” This allegation should be supported by references. I think that at the very least in a UK context the NICE guideline NG143 should be mentioned. Methods page 5 Line 12-14 “We used previous research [16, 17] to develop topic guides and participant information sheets (PIS).” This could be briefly described. Line 23-24 SMG: first occurrence, please develop the acronym. Line 23-25 “They provided valuable input into the design and conduct of the study,” According to the context of the paper the word “valuable” is already a conclusion in itself, I would suggest reporting the input in the results section and the qualification of “valuable” in the discussion part. It would have been interesting to know if participants had received any information referring to a standardized method of temperature measurement (site and device). Results page 10 This part could be improved if the authors were able to propose a consort flow diagram for the participation (parents and staff) covering the 3 stages. For the same purpose the participants and non-participants characteristics could be described in a table. parents: age; gender, diploma, socio-professional category, underlying conditions of the children staff: age, gender, position, experience in PICU (years) Are the participants different from non-participants? Page 14 “Interestingly, staff trained by their local unit colleagues were significantly more likely to find the permissive threshold not acceptable when compared to those trained directly by the pilot trial team ($X^2(2) = 8.78, p = 0.012$).” As mentioned previously it would be interesting to explore if there is any difference between both groups according to (age, gender, position, experience in PICU (years))
--	--

REVIEWER	Arti Maria ABVIMS & Dr RML Hospital New Delhi India
REVIEW RETURNED	24-Sep-2020

GENERAL COMMENTS	Detailed Page wise Comments Overall:
---

	1. The writing is not reader friendly, 2. The methodology is too patchy (needs much detailing) and 3. The tables are jumbled Title: 1. Mentions as an Acceptability study; but later in abstract n text they mention it as a feasibility study? In abstract 1. Line 5-6..How have the perspectives been augmented? The meaning of the sentence is not clear 2. Line 23...“yet some concerns regarding proposed temperature thresholds and not using paracetamol for pain or discomfort”: Please consider to simplify the sentence. 3. Line 46..In article summary : a. Interviews and focus groups are methods. Questionnaire is a tool. It cannot be mentioned as a type of method. Can be termed as surveys. Main Manuscript Introduction 1. Line 15...“eligible population is smaller” .. replace by younger. 2. Line 57.. Simplify Methodology The methods section is cryptic and difficult to follow. The authors have directed the reader to their other publications - perhaps, those may have more clarity. However, since some readers may not want to refer to articles elsewhere, it is advisable that the authors give a background to what they really did. 1. Spell errors a. Line 20..Parent not patent... b. Line 40...Advertising not adverting 2. Line 37..What is the definition of children? what age bracket has been considered? 3. Line 46.. any other eligibility criteria apart from what has been mentioned in lines 37-38? 4. Elaborate a. Line 53 ...relevant staff..... Who?? b. Line 58...voting... How?? 5. Line 56.. Close-ended and not 'closed' questions! Pilot RCT: Concurrent and retrospective recruitment and conduct 1. Line 14...This needs clarity. Why were consent questionnaires given out before discharge? it is unclear what the FEVER study is - is it a hospital-based RCT or is it something else. Suggest the authors include a box that summarizes the details of the FEVER RCT even if they have guided the reader to other articles with details. 2. Line 17...What is the rationale of giving a month's buffer time between 'concurrent' and 'retrospective' recruitment? Why did the team need two types of recruitment? 3. Line 17-18.. Again a potential situation for self-selection. Do we have any profiling to verify? 4. Line 19..Sequential order needs clarification. What was the order? 5. Line 25.... Focus group needs details. Why was it conducted? How was it conducted? What was the composition of each of the group? Was it mixed ? 6. Line 53...The representation in the tables is inconsistent. Suggest: Remove the quotes and provide only the codes. Keep a sequence for qualitative and quantitative information. Can provide the codes first and then the quantitative data (or the reverse,
--	---

	whichever improves readability) - in the current form, quantitative information has been provided against qualitative data collection methods - needs correction. Similarly, Table 2 provides quotable quotes while the row suggests that data was collected using survey questionnaire - again, needs correction. 7. Table 1 page 6, a. Line 3.. It is better to simplify and present the table in terms of enablers and barriers , using the codes, rather mixing the quantitative, qualitative and the quotable quotes together. It makes understanding cumbersome. b. Must provide list of abbreviations in the Table legend/ footnotes c. Line16.. The methodology needs to clarify how the participants have been coded. Both the parents and the staff are coded as P? d. line 22...How have we calculated 82%? Has the groups been considered or the participants only? Needs to be clarified. 8. Table 2 Page 7 .. a. line 3...title of table 2: Can please explain what this means. b. Line 15-28..The table needs to mention why certain portions of the tables are highlighted? Some index needs to be there. 9. Page 8 Table 3 a. Title: Retrospective not respective!! 10. Page 9 Results: a. Line..11..Can mention that some leniency has been undertaken to go beyond 3 years (36 months) b. Line 16.. can mention that some leniency has been undertaken to go beyond 3 years (36 months) c. Line 22.. how many singles, twins and triplets? some details need to be mentioned d. Line 24..."48 from 47 families".... ABSURD??? e. Line 36.. retrospective has been allowed for more than 30 days? 11. Page 10 a. Line7...This quote also appears in Table 1 under 'perceived effectiveness'. Can avoid duplication 12. Page 13 a. Line 52 The word "intransigently" seems to have been used out of place. Can replace/ simplify the word. 13. Page 14, line 12, How was it ensured that 'all key stakeholders' had been included? 14. Figure 3: page 22, ..Line 22. Is there any reason why this adapted framework has the types of acceptability listed in the vertical form (which is different from the original)? 15. Participant consent sketchily addressed. 16. Outcomes are not very clear..some place mention acceptability and at other places as feasibility study 17. References : Probably all do not seem completeand there is a need to recheck.. for eg. 1, 4,7, 13,16. 21, 25 & 26
--	---

REVIEWER	Heidi Holmen Oslo Metropolitan University, Norway
REVIEW RETURNED	26-Nov-2020

GENERAL COMMENTS	Dear editor and authors, Thank you for the opportunity to review the current manuscript on the acceptability of pediatric critical care intervention. The manuscript presents an important topic and overall, the research aim is well justified and there definitely is a need for research with a broader view on acceptability as the researchers claim, and not only within pediatric critical care. My general impression is good, however there are a lack of details and some clarifications that
--

would improve the manuscript and increase its readability. I would also encourage the authors to include a section to describe how this research is useful for future research more explicitly. I would regard this paper to have an impact and be of interest to a wider group than pediatric care, and I would urge the authors to choose sound keywords in order to reach their potential audience.

Some specific comments:

1. Would it be possible to describe the intervention in one sentence in the abstract to clarify the study purpose?
2. The introduction is clear and to the point, setting out a clear context and need for the current research. I would however suggest adding some more details regarding the overall pilot RCT to increase the understanding of the findings from this acceptability study. For example, details on the intervention, inclusion criteria to be included, follow-up time and measures performed. This would make it easier to understand the relevance of the timepoints from interview, e.g. one months after randomization – this doesn't give much sense if we do not know of the trial lasts for weeks or months. The one-month perspective would be different in a three-week study compared to a three month trial. It is good the other studies are referred to, but this particular manuscript should be understandable on its own.
3. Page 4 of the pdf, line 59, I guess there's a type, and the word is (...) through the lens of (...)?
4. There is an excess use of abbreviations, one which remain unexplained on page 5, line 23 – please reduce the number and add them all in an explanatory text if they are crucial for your paper.
5. Page 4, line 45 I believe this sentence needs revisions, possible add "and" after the last comma? A draft pilot RCT PIS was emailed to parents prior to interview, which took place with ED in person or via telephone based on parent preference, consent was obtained.
6. In the methods, under design, as I read through the paper I get the impression that interviews and data were gathered pre, per, and post – is that correct, and should it be stated that in this mixed methods study, data were also collected post-trial to add retrospective perspectives on the acceptability? Further, the very first sentence under design could be revised to be a complete sentence, likewise the last sentence. This section has a lack of clear language and lack of details. For example, the authors state that they used previous research to develop "topic guides" and I wonder if they could explain what these topics guided – e.g. the interviews or the analysis? Other? Please elaborate.
7. Heading on PPI should be "Patient and public involvement?"
8. Staff focus groups, page 5; It remain unclear whether the authors suggest that the closed questions can be characterized as focus groups, or whether the healthcare personnel group are participants of a former focus group – please revise for clarity. This goes for the next page and the repeated interviews/ survey.
9. Page 6, line 19, please add some details on the consequences the stratification had for the invitations to interview – did the researchers invite one from each stratifies arm every other time? "stratifying by study arm (lower/higher temperature threshold)."
10. Please consider adding a section for the ethical perspectives, although approval is stated after the main body text.
11. Methods under analysis, I suggest revising the statistics as chi square is a descriptive statistical analysis, and if only descriptive statistics were applied you might state that, otherwise describe

	how you analyzed both categorical and continuous data. I would also urge the authors to include more details on their thematic analysis – how many researchers were involved in the iterative process? Where the users involved in these? How were the material coded, etc. Add details for the possibility of assessing the work and the soundness of the method. 12. Page 10, line 15, again a missing “r” in what I guess is supposed to be “through”. 13. Gender perspectives – as in research in this field, it is common to see more mothers engaged in research compared to father. I see quotes from both genders, but in the post-trial section of the results there are only quotes from fathers. Any reason or coincidence? And just on a side note – any reason why the gender of the researchers are given in the text? 14. Table 1, 2, and 3, please add text to explain why some fields are shaded and others are not – I cannot find this information in the text? 15. The reference list contains several typos and excess characters, please revise accordingly.
--	---

VERSION 1 – AUTHOR RESPONSE

Reviewer: 1

Reviewer Name: Fabrice Ruiz

Institution and Country: ClinSearch France

Please state any competing interests or state ‘None declared’: None declared

Comments to the Author

This article addresses the critical issue of the acceptability of pediatric RCT to parents and health care professionals based on the development of the protocol and appendices (eg: Participant Information Sheets) and the pilot of the FEVER study (protocol for temperature management in critically ill children with fever and infection). Study findings have already been published, and this paper tends to focus on parent and staff perspectives on trial acceptability.

General comment

Upon reading the abstract, we can thought that the subject of fever management had been chosen by the researchers to serve as a basis for demonstrating the characteristics of their method to increase the preparedness of patients/parents and healthcare professionals for RCT to establish and augment their acceptability.

As such, I would suggest clarifying this point in the abstract and introducing it earlier in the introduction section.

Response: we have amended the abstract to help clarify the objective of the study. We have added the full name of the proposed trial (the Fever trial) to help clarify that this was not a chosen subject but a feasibility study to inform the design of a randomised control trial. We have also added to the design to further clarify that we are referring to data collected at three time points during the feasibility study. We hope these amendments and the new Figure 2 in the methods section help to clarify our focus and study design

The authors have certainly put a lot of effort into these investigations, which in my opinion deserve an elaborate illustration of the design in the section “Methods”. This could guide the reader to approach the paper more comfortably.

Response: we have added an additional figure to cover the design of the FEVER trail and the feasibility study, highlighting the elements that are not reported in this paper to help clarify the methods we discuss. We have also amended the methods study design section.

Introduction page 4

The first part of the introduction could be enriched with some more recent references: the authors will be able to rely on a document drafted by the Enpr-EMA which has just been edited and available via the following link: https://www.ema.europa.eu/en/documents/other/preparedness-medicines-clinical-trials-paediatrics-recommendations-enpr-ema-working-group-trial_en.pdf

Response: thank you for drawing our attention to this useful document, we have updated our references.

Line 32-33 “To our knowledge, the TFA has not been used in the analysis of real world data”. This sentence should be reworded because there are now a few references in recent literature.

Response: this sentence was correct at the time of writing but has now been deleted. Thank you.

Line 38-40 “There is a lack of robust evidence or guidance to inform the management of fever due to an infection in critically ill children.” This allegation should be supported by references. I think that at the very least in a UK context the NICE guideline NG143 should be mentioned.

Response: we have added the NICE guidelines and the Brick et al 2016 reference to support this statement.

Methods page 5

Line 12-14 “We used previous research [16, 17] to develop topic guides and participant information sheets (PIS).” This could be briefly described.

Response: we have expanded and added an additional reference “We used previous research on patient and staff perspectives on trials conducted in paediatric emergency and critical care in the NHS”

Line 23-24 SMG: first occurrence, please develop the acronym.

Response: replaced with Study Management Group as this is the only time we refer to it in the manuscript

Line 23-25 “They provided valuable input into the design and conduct of the study,” According to the context of the paper the word “valuable” is already a conclusion in itself, I would suggest reporting the input in the results section and the qualification of “valuable” in the discussion part.

Response: this section is about PPI and not study participants therefore placement in the results would not be appropriate. PPI input into the design of the study as members of our study team was extremely valuable and we would like to keep this acknowledgement if possible.

It would have been interesting to know if participants had received any information referring to a standardized method of temperature measurement (site and device).

Response: we did not specify a standardized method of temperature measurement in the pilot trial

protocol so this information was not provided.

Results page 10

This part could be improved if the authors were able to propose a consort flow diagram for the participation (parents and staff) covering the 3 stages.

Response: This is provided in Figure 3 (originally figure 2) which provides detail of participant characteristics for each of the three study stages.

For the same purpose the participants and non-participants characteristics could be described in a table. Parents: age; gender, diploma, socio-professional category, underlying conditions of the children staff: age, gender, position, experience in PICU (years)

Are the participants different from non-participants?

Response: it was not possible to collect such data from parents or staff who did not provide consent for participation in the study. They would have needed to participate to gain such information. An important participant characteristic for this particular study was ensuring the inclusion of parents who declined their child's participation in the pilot RCT as insight into their reasons for declining were crucial in assessing trial acceptability, and indeed, overall feasibility. We have added to the study discussion to describe how: Insight was gained into the views of 8 (2 interviews 6 questionnaires) out of 18 parents (44%) who had declined their child's continued participation in one or more aspect of the pilot RCT. In particular, the interviews with parents who declined consent and nursing staff who found the protocol challenging to follow provided valuable information to assist with refining the study process for a definitive RCT. However, it is unknown whether or not the predominantly positive views of the declining parents who took part in an interview or questionnaire were shared by other parents who declined the pilot RCT.

Page 14 "Interestingly, staff trained by their local unit colleagues were significantly more likely to find the permissive threshold not acceptable when compared to those trained directly by the pilot trial team ($X^2(2) = 8.78, p = 0.012$)."

As mentioned previously it would be interesting to explore if there is any difference between both groups according to (age, gender, position, experience in PICU (years))

Response: we agree this was an interesting findings and one that has changed how site staff training is delivered through the ICNARC trials unit. Of the characteristics suggested we only had data on position and there were no notable differences to report.

Reviewer: 2

Reviewer Name: Arti Maria

Institution and Country: ABVIMS & Dr RML Hospital, New Delhi, India

Please state any competing interests or state 'None declared': None

Comments to the Author

Detailed Page wise Comments

Overall:

1. The writing is not reader friendly,

Response: we have amended any sections of the paper that have been raised as unclear with the aim of making it more reader friendly.

2. The methodology is too patchy (needs much detailing) and

Response: additional information has been given alongside a new methods figure for clarity. We have also added in a reference to the full NIHR HTA monograph which provides further details of study methods.

3. The tables are jumbled

Response: please see response linked to 'The representation in the tables is inconsistent' (methodology number 6)

Title:

1. Mentions as an Acceptability study; but later in abstract n text they mention it as a feasibility study?

Response: the paper explores views on trial acceptability within a wider feasibility study. We have aimed to clarify this within the revised study design section.

In abstract

1. Line 5-6..How have the perspectives been augmented? The meaning of the sentence is not clear
Response: this has been amended.

2. Line 23..."yet some concerns regarding proposed temperature thresholds and not using paracetamol for pain or discomfort": Please consider to simplify the sentence.

Response: this has been amended to help clarify.

3. Line 46..In article summary : Interviews and focus groups are methods. Questionnaire is a tool. It cannot be mentioned as a type of method. Can be termed as surveys.

Response: we have changed the word questionnaire for survey

Main Manuscript

Introduction

1. Line 15..."eligible population is smaller" .. replace by younger.

Response: this would not be accurate s as this relates to the population size not the size/age of the child.

2. Line 57.. Simplify

Response: this has been simplified

Methodology

The methods section is cryptic and difficult to follow. The authors have directed the reader to their other publications - perhaps, those may have more clarity. However, since some readers may not want to refer to articles elsewhere, it is advisable that the authors give a background to what they really did.

Response : we have added additional detail around the conduct of the concurrent and retrospective conduct (see following points) and a figure that should improve clarity

1. Spell errors

a. Line 20..Parent not patent...

response: patient not patent, this was incorrect in three places in the manuscript and has been changed

b. Line 40...Advertising not adverting

Response: this has been changed

2. Line 37..What is the definition of children? what age bracket has been considered?

Response: we did not define child in our inclusion criteria as they would only be admitted to a paediatric intensive care unit if they were children (defined in the UK as under 16 years of age). We have added the age bracket for clarity for non-UK readers

3. Line 46.. any other eligibility criteria apart from what has been mentioned in lines 37-38?

Elaborate

Response: participants also needed to speak English, this has been added

a. Line 53 ...relevant staff..... Who??

Response : replaced with "all staff who would be involved in the conduct of a clinical trial within a paediatric intensive care unit "

b. Line 58...voting... How??

Response : this had been clarified by restating 'the Turning Technologies (Youngstown, OH, USA) voting system.'

5. Line 56.. Close-ended and not 'closed' questions!

Response : changed in two places to closed ended.

Pilot RCT: Concurrent and retrospective recruitment and conduct

1. Line 14...This needs clarity. Why were consent questionnaires given out before discharge?

Response: This was the maximum timeframe to increase the likelihood of completion. In reality, parents completed it following the recruitment discussion. We have amended to more accurately reflect what happened.

it is unclear what the FEVER study is - is it a hospital-based RCT or is it something else. Suggest the authors include a box that summarizes the details of the FEVER RCT even if they have guided the reader to other articles with details.

Response: we have added that the FEVER study is 'hospital-based' in the introduction and a new figure summarising the FEVER Study and FEVER RCT has been included in the methods.

2. Line 17...What is the rationale of giving a month's buffer time between 'concurrent' and 'retrospective' recruitment? Why did the team need two types of recruitment?

Response: it would have been inappropriate to conduct time consuming interviews when children were critically ill. We arranged interviews when families were back at home and children had recovered.

3. Line 17-18.. Again a potential situation for self-selection. Do we have any profiling to verify?

Response: please see our response to reviewer 1 on this issue and new section added to the discussion to acknowledge sample limitations.

4. Line 19..Sequential order needs clarification. What was the order?

Response: in order of receipt of a consent form. This has been clarified.

5. Line 25.... Focus group needs details. Why was it conducted? How was it conducted? What was the composition of each of the group? Was it mixed ?

Response: additional information has been added. Sample characteristics are in the results section and Figure 3.

6. Line 53...The representation in the tables is inconsistent. Suggest: Remove the quotes and provide only the codes. Keep a sequence for qualitative and quantitative information. Can provide the codes first and then the quantitative data (or the reverse, whichever improves readability) - in the current form, quantitative information has been provided against qualitative data collection methods - needs correction.

Response: the current format reflects the mixed methods framework analysis conducted. We would be concerned that removing quotations would remove justification for the findings, make it difficult for the reader to distinguish between qualitative, quantitative data and the brief statements, which were derived from both multiple datasets. We hope the reviewer does not mind if we keep the Table as it is.

Similarly, Table 2 provides quotable quotes while the row suggests that data was collected using survey questionnaire - again, needs correction.

Response: the questionnaire included open ended/ free box answers and therefore these have been quoted. This has been explained in the methods section to help the reader understand the data source.

7. Table 1 page 6,

Line 3.. It is better to simplify and present the table in terms of enablers and barriers, using the codes, rather mixing the quantitative, qualitative and the quotable quotes together. It makes understanding cumbersome.

Response: As described above, the table reflects the mixed methods framework analysis that was conducted using the TFA. Our analysis was not focussed on a more simple barriers and enablers analysis therefore we do not feel it would be appropriate to change it as suggested. We hope the reviewer understands.

c. Must provide list of abbreviations in the Table legend/ footnotes

d. Response: A key has been added for each table explaining the abbreviations

e. Line16.. The methodology needs to clarify how the participants have been coded. Both the parents and the staff are coded as P?

Response: P stands for participant. This has been clarified in the methods. We have also added the word 'staff' to the practitioners identifiers to add clarity between the two groups.

f. line 22...How have we calculated 82%? Has the groups been considered or the participants only? Needs to be clarified.

Response : clarified in table "82 % (45/55, one missing)

8. Table 2 Page 7 ..

a. line 3...title of table 2: Can please explain what this means.

Response : spelling error and had been addressed

. Thank you

b. Line 15-28..The table needs to mention why certain portions of the tables are highlighted?

Some index needs to be there.

Response: A key has been added for each table explaining the highlights

9. Page 8 Table 3

a. Title: Retrospective not respective!!

Response: this has been changed. Thank you.

10. Page 9 Results:

a. Line..11..Can mention that some leniency has been undertaken to go beyond 3 years (36 months)

Response: we have added this to the methods section

b. Line 16.. can mention that some leniency has been undertaken to go beyond 3 years (36 months)

Response: please see above

c. Line 22.. how many singles, twins and triplets? some details need to be mentioned

Response : as our participants were parents not their children this detail was not included in the participant characteristics section. We do not have access to such data for pilot trial participants or questionnaire participants. No parents discussed having twin or triplets in the fever pilot trial so this was not something that appeared to impact on decision-making but we do not feel it is appropriate to add this to the paper due to a lack of supporting data.

d. Line 24..."48 from 47 families".... ABSURD???

Response: Recruiters asked both parents to fill in the questionnaire (this had been added to the methods section). In most cases only one did. Two parents completed the questionnaire from one family. This is correct.

g. Line 36.. retrospective has been allowed for more than 30 days?

Response: parents were contacted within 30 days not always interviewed within 30 days. This has

been checked- the methods does state approximately one month.

11. Page 10

a. Line7...This quote also appears in Table 1 under 'perceived effectiveness'. Can avoid duplication
Response: we would like to keep this quote in the text and Table if possible.

12. Page 13

a. Line 52 The word “intransigently” seems to have been used out of place. Can replace/ simplify the word.

Response: amended. Thank you

13. Page 14, line 12, How was it ensured that 'all key stakeholders' had been included?

Response: we made sure we had included both parents with relevant experience and staff from the trial sites- who were the key stakeholders for this study. Removed 'all'

14. Figure 3: page 22, ..Line 22. Is there any reason why this adapted framework has the types of acceptability listed in the vertical form (which is different from the original)?

Response: We did try the alternative format but due to the addition of an additional theme the figure is clearer presented this way.

15. Participant consent sketchily addressed.

Response: we have added more detail how on consent was sought for both parents and clinicians.

16. Outcomes are not very clear... some place mention acceptability and at other places as feasibility study

Response: we hope the acceptability element of the wider feasibility study is now clear for the reader with the amendments made.

17. References : Probably all do not seem complete and there is a need to recheck.. for eg. 1, 4,7, 13,16. 21, 25 & 26

Response: these have all been reviewed and amended

Reviewer: 3

Reviewer Name: Heidi Holmen

Institution and Country: Oslo Metropolitan University, Norway

Please state any competing interests or state 'None declared': None declared.

Comments to the Author

Dear editor and authors,

Thank you for the opportunity to review the current manuscript on the acceptability of pediatric critical care intervention. The manuscript presents an important topic and overall, the research aim is well justified and there definitely is a need for research with a broader view on acceptability as the researchers claim, and not only within pediatric critical care. My general impression is good, however there are a lack of details and some clarifications that would improve the manuscript and increase its readability. I would also encourage the authors to include a section to describe how this research is useful for future research more explicitly.

I would regard this paper to have an impact and be of interest to a wider group than pediatric care, and I would urge the authors to choose sound keywords in order to reach their potential audience.

Response: thank you for your comments, we have reviewed the key words ad added 'practitioner training '

Some specific comments:

1. Would it be possible to describe the intervention in one sentence in the abstract to clarify the study purpose?

Response: this has been added (antipyretic intervention). Thankyou

2. The introduction is clear and to the point, setting out a clear context and need for the current research. I would however suggest adding some more details regarding the overall pilot RCT to

increase the understanding of the findings from this acceptability study. For example, details on the intervention, inclusion criteria to be included, follow-up time and measures performed. This would make it easier to understand the relevance of the timepoints from interview, e.g. one month after randomization – this doesn't give much sense if we do not know of the trial lasts for weeks or months. The one-month perspective would be different in a three-week study compared to a three month trial. It is good the other studies are referred to, but this particular manuscript should be understandable on its own

Response : we agree more clarity is needed- as the design of the study changed during the process additional information has been added within the methods rather than the introduction and as part of the new figure. The methods section now includes: The pilot RCT took place over a 3 month period (October to December 2017). Children were randomly allocated (1 : 1) using research without prior consent (RWPC) to permissive (39.5 °C) or restrictive (37.5 °C) temperature thresholds for antipyretics during their PICU stay while mechanically ventilated.

3. Page 4 of the pdf, line 59, I guess there's a type, and the word is (...) through the lens of (...)?

Response: this has been amended thankyou

4. There is an excess use of abbreviations, one which remain unexplained on page 5, line 23 – please reduce the number and add them all in an explanatory text if they are crucial for your paper.

Response: this abbreviation has been removed and keys have been added to the results tables for clarity of reading.

5. Page 4, line 45 I believe this sentence needs revisions, possible add “and” after the last comma? A draft pilot RCT PIS was emailed to parents prior to interview, which took place with ED in person or via telephone based on parent preference, consent was obtained.

Response: added. thankyou

6. In the methods, under design, as I read through the paper I get the impression that interviews and data were gathered pre, per, and post – is that correct, and should it be stated that in this mixed methods study, data were also collected post-trial to add retrospective perspectives on the acceptability? Further, the very first sentence under design could be revised to be a complete sentence, likewise the last sentence. This section has a lack of clear language and lack of details. For example, the authors state that they used previous research to develop “topic guides” and I wonder if they could explain what these topics guided – e.g. the interviews or the analysis? Other? Please elaborate.

Response: This has been amended thankyou.

7. Heading on PPI should be “Patient and public involvement?”

Response: this has been changed.

8. Staff focus groups, page 5; It remain unclear whether the authors suggest that the closed questions can be characterized as focus groups, or whether the healthcare personnel group are participants of a former focus group – please revise for clarity. This goes for the next page and the repeated interviews/ survey.

Response: we have added additional information about the focus group conduct to clarify that the topic guide included a mix of open and close ended questions and our justification for doing so. We have also added some examples of how the Turning Point administered closed questions were used to help facilitate group discussion. We hope this is now clearer for the reader.

9. Page 6, line 19, please add some details on the consequences the stratification had for the invitations to interview – did the researchers invite one from each stratifies arm every other time?

“stratifying by study arm (lower/higher temperature threshold).

Response: We have added to this section to clarify: “stratifying by study arm (lower/higher temperature threshold) as the study progressed ensuring parents whose children had been randomised to both trial arms were represented in the sample”. It was not every other time as it does not work out that clear cut in practice with who consents and who responds to request or arranged interview times. We hope this helps to clarify.

10. Please consider adding a section for the ethical perspectives, although approval is stated after the main body text.

Response: We are a little unclear as to what specific section the reviewer is referring to. As mentioned we have included how research ethics approval was obtained and additional information on how consent was obtained. If further details of this process are required by the editor please let us know what is required.

11. Methods under analysis, I suggest revising the statistics as chi square is a descriptive statistical analysis, and if only descriptive statistics were applied you might state that, otherwise describe how you analyzed both categorical and continuous data.

Response: we have removed ‘chi squared’ and stated descriptive analysis.

I would also urge the authors to include more details on their thematic analysis – how many researchers were involved in the iterative process? Where the users involved in these? How were the material coded, etc. Add details for the possibility of assessing the work and the soundness of the method.

Response: we have added an additional table outlining the thematic analysis process.

12. Page 10, line 15, again a missing “r” in what I guess is supposed to be “through”.

Response: this has been amended. Thank you.

13. Gender perspectives – as in research in this field, it is common to see more mothers engaged in research compared to father. I see quotes from both genders, but in the post-trial section of the results there are only quotes from fathers. Any reason or coincidence?

Response: this is purely coincidental, in this case the fathers had the most illustrative quotes. We have checked the corresponding results table and there is a mix of mothers and fathers represented.

And just on a side note – any reason why the gender of the researchers are given in the text?

Response: this was included because the COREQ checklist under personal characteristics requests the gender of the researchers.

14. Table 1, 2, and 3, please add text to explain why some fields are shaded and others are not – I cannot find this information in the text?

Response: ‘shaded fields highlight potentially unacceptable aspects of the trial’ this has been added as part of the new table Keys.

15. The reference list contains several typos and excess characters, please revise accordingly.

Response: All references have been reviewed.

VERSION 2 – REVIEW

REVIEWER	Fabrice RUIZ ClinSearch France
REVIEW RETURNED	27-Jan-2021

GENERAL COMMENTS	thank you to the authors for considering and responding favourably to most of the reviewers' comments
REVIEWER	Arti Maria ABVIMS & Dr. Ram Manohar Lohia Hospital. New Delhi India
REVIEW RETURNED	28-Jan-2021
GENERAL COMMENTS	I think most of the concerns raised previously have been addressed. However some typo errors need to be corrected.
REVIEWER	Heidi Holmen Oslo Metropolitan University, Norway
REVIEW RETURNED	26-Jan-2021
GENERAL COMMENTS	Thank you for the good response to our inquires, the paper reads well now. some minor errors, easily addressed. My comments are based on the document WITH track changes. Page 26, line 46: "Comparing a" - is written twice. Page 30 - consider using the same objective in the introduction as in the abstract for consistency. page 30 - first you delete the abbreviation for RWPC, then you re-introduce. I say delete, this is not a a standard one? page 30-31 - be consistent in your abbreviations - PIS is somewhere used and others not. Page 31 line 46: Turning Point is introduced, and i guess this refers to the voting system, but please revise for clarity. Reuse the sentence on page 32, line 38 which was clear. My previous comment on ethics can be ignored, as you have provided sufficient details.

VERSION 2 – AUTHOR RESPONSE

Reviewer: 1

Dr. Fabrice Ruiz, ClinSearch -110

Comments to the Author:

thank you to the authors for considering and responding favourably to most of the reviewers' comments

Response: We agree this version of the paper is much clearer. Thank you for your help improving it.

Reviewer: 2

Dr. Arti Maria, ABVIMS & Dr RML Hospital

Comments to the Author:

I think most of the concerns raised previously have been addressed.

Response: We agree this version of the paper is much clearer. Thank you for your help improving it. However some typo errors need to be corrected.

Response: The paper has been proof read and typos addressed.

Reviewer: 3

Dr. Heidi Holmen, Oslo Metropolitan University Comments to the Author:

Thank you for the good response to our inquires, the paper reads well now.

Response: We agree this version of the paper is much clearer. Thank you for your help improving it. some minor errors, easily addressed. My comments are based on the document WITH track changes.

Response: as these errors are also present on the 'clean' copy, for clarity that version has been

amended creating a new 'Marked' copy.

Page 26, line 46: "Comparing a" - is written twice.

Response: repetition removed

Page 30 - consider using the same objective in the introduction as in the abstract for consistency.

Response: We have cross checked the wording and adjusted both for consistency.

page 30 - first you delete the abbreviation for RWPC, then you re-introduce. I say delete, this is not a standard one?

Response: you are correct that it should not have been deleted in the first instance. We have moved it earlier and removed later. We think the abbreviation is needed as RWPC is mentioned 19 times throughout the manuscript and is too long to write in full each time.

page 30-31 - be consistent in your abbreviations - PIS is somewhere used and others not.

Response: thank you for highlighting this we have changed two 'participant information sheets' to PIS. In addition we noticed PIS was missing from the abbreviation list for table 2. This has now been added.

Page 31 line 46: Turning Point is introduced, and i guess this refers to the voting system, but please revise for clarity. Reuse the sentence on page 32, line 38 which was clear.

Response: Amended using the sentence on page 32

My previous comment on ethics can be ignored, as you have provided sufficient details.

Response: Thank you